# Evaluating the Effectiveness of Water-Saving Irrigation on Wheat (*Triticum aestivum* L.) Production in China: A Meta-Analytical Approach

**DOI:** 10.3390/plants14182837

**Published:** 2025-09-11

**Authors:** Jiayu Ma, Baozhong Yin, Cuijiao Jing, Wanyi Li, Yilan Qiao, Luyao Zhang, Haotian Fan, Limin Gu, Wenchao Zhen

**Affiliations:** 1State Key Laboratory of North China Crop Improvement and Regulation/Key Laboratory of North China Water-Saving Agriculture, Ministry of Agriculture and Rural Affairs/Key Laboratory of Crop Growth Regulation of Hebei Province, College of Agronomy, Hebei Agricultural University, Baoding 071001, China; 13673148166@163.com (J.M.); 17332317940@163.com (C.J.); 15188603417@163.com (W.L.); 15130609159@163.com (Y.Q.); 19833895188@163.com (L.Z.); 13091874885@163.com (H.F.); 2College of Plant Protection, Hebei Agricultural University, Baoding 071001, China; yinbaozhong@hebau.edu.cn

**Keywords:** wheat, water-saving irrigation, water use efficiency, yield, meta-analysis

## Abstract

Optimized water-saving irrigation (WSI) practices are critical for enhancing resource use efficiency and ensuring sustainable wheat production in water-scarce regions. This meta-analysis quantitatively assessed the effects of various WSI methods on wheat yield, water use efficiency (WUE), and partial factor productivity of nitrogen (PFPN) across China’s wheat regions. The results showed that optimized irrigation, particularly drip and micro-sprinkler systems, significantly reduced irrigation water and nitrogen inputs by 35.1% and 7.2%, respectively, without yield penalties. Drip and micro-sprinkler irrigation, which together accounted for over 97% of observations, improved WUE by 18.7% and 10.1%, respectively, and increased PFPN by 6.8% and 5.5%, highlighting their dominant role in current WSI practices. Moderate deficit irrigation (60–100% of full irrigation) optimized WUE and PFPN while maintaining stable yields, whereas severe deficit irrigation (<40%) caused substantial yield losses. Soil texture and bulk density strongly modulated WSI effectiveness. Climatic factors, particularly growing season precipitation, negatively correlated with WSI benefits, highlighting enhanced efficiency gains under drier conditions. These findings emphasize the need to prioritize drip and micro-sprinkler irrigation in national water-saving strategies and advocate for integrated approaches combining WSI with soil health management and site-specific irrigation scheduling to promote sustainable wheat intensification under variable agroecological conditions.

## 1. Introduction

Global agriculture is under mounting pressure to meet increasing food demand while conserving key natural resources such as water and nitrogen [1,2]. Wheat (*Triticum aestivum* L.), one of the most widely cultivated staple crops worldwide, is grown extensively in water-limited regions, where irrigation plays a crucial role in sustaining yield and ensuring food security [3]. However, excessive use of irrigation water and nitrogen fertilizers has led to severe inefficiencies and environmental degradation, including water scarcity, nutrient leaching, and greenhouse gas emissions [4,5,6]. As a result, the development and implementation of efficient water and nutrient management strategies have become central goals for sustainable wheat production systems [1,7].

Water-saving irrigation (WSI) practices—such as drip irrigation, micro-sprinkler irrigation, alternate root-zone irrigation, and regulated deficit irrigation—are increasingly promoted as promising approaches to enhance water use efficiency (WUE), reduce nitrogen losses, and maintain or even improve crop yields [8]. These techniques aim to deliver water more precisely in both spatial and temporal dimensions, minimizing evaporation and percolation losses and synchronizing water and nitrogen availability with crop demand [9]. Notably, drip and micro-sprinkler systems, which provide high-frequency, low-volume irrigation, have been shown to enhance root development and nutrient uptake, reduce deep percolation, and stabilize yield under water-limited conditions [10].

In wheat production systems across China, WSI practices have been widely studied for their potential to reduce irrigation water use (IW), sustain or increase yield, and improve WUE [11,12,13]. While several studies have reported that WSI can significantly reduce water input without compromising yield [14,15], others have found that WSI may lead to yield reductions [16,17]. The yield response to WSI remains inconsistent, and many researchers emphasize that the variability of results depends on environmental and management conditions [18,19]. Nevertheless, most field experiments are constrained by site-specific conditions and limited spatial scales, restricting the generalizability of their findings. Differences in soil properties, wheat phenology, and regional climates have contributed to the heterogeneity in reported results. Therefore, a clear consensus on the overall effectiveness of WSI under diverse environments is still lacking. Understanding how WSI practices interact with environmental and climatic variables is essential for informing site-specific irrigation strategies. To address these knowledge gaps, meta-analysis provides a robust quantitative method to synthesize data across field studies, identify generalizable trends, and assess the magnitude and consistency of WSI effects on wheat systems. Moreover, although nitrogen plays a critical role in determining crop productivity and resource use efficiency, the combined effects of WSI and nitrogen management remain underexplored.

In this study, we conducted a comprehensive meta-analysis of over 57 peer-reviewed field experiments to compare the impacts of WSI practices with conventional irrigation methods (typically full surface or furrow irrigation) in wheat production systems. Water and nitrogen are tightly coupled in wheat systems through transport, transformation, and uptake processes. Irrigation controls nitrogen movement to roots via mass flow and diffusion, modulates soil mineralization and the balance between nitrification and denitrification, and indirectly affects nitrogen use through stomatal regulation and canopy conductance [20,21]. High-frequency micro-irrigation (e.g., drip, micro-sprinkler) can improve water–nitrogen synchrony by reducing percolation losses and collocating moisture and nutrients in the root zone, whereas excessive deficits risk decoupling nitrogen supply from crop demand [22,23]. Consequently, assessing water-saving irrigation (WSI) solely with water-centric indicators may understate its agronomic value. In this study we therefore evaluate WSI effects not only on yield and water use efficiency (WUE) but also on nitrogen productivity (PFPN), and we examine how irrigation levels and site factors condition these water–nutrient synergies.

The specific objectives were to: (1) quantify the overall effects of different WSI strategies—particularly drip and micro-sprinkler irrigation—on yield, WUE, and partial factor productivity of nitrogen (PFPN); (2) assess how these effects vary across irrigation levels, soil bulk density, and soil texture; and (3) examine the relationships between climatic variables—namely growing season precipitation (GSP) and mean annual temperature (MAT)—and agronomic outcomes using meta-regression analysis.

## 2. Results

### 2.1. Effects of Optimized Irrigation on Water and Nitrogen Input Levels

The OWN treatment, which represents optimized irrigation methods such as drip and micro-sprinkler irrigation, significantly reduced both irrigation water and nitrogen fertilizer inputs compared with the conventional treatment (CK) across studies included in the meta-analysis (Figure 1a). On average, irrigation amount under OWN was 35.1% lower than that under CK (*p* < 0.0001, Figure 1a), demonstrating substantial water savings. Similarly, nitrogen application rate under OWN was 7.2% lower than that under CK (*p* < 0.01, Figure 1b).

### 2.2. Effects of Different Water-Saving Irrigation Methods on Wheat Yield, WUE, and Partial Factor Productivity of Nitrogen

Among the various water-saving irrigation methods, micro-sprinkler irrigation significantly increased wheat yield by an average of 4.1%, whereas drip irrigation did not result in a statistically significant yield response (Figure 2a). Both drip and micro-sprinkler irrigation substantially improved water use efficiency (WUE), with average increases of 18.7% and 10.1%, respectively (Figure 2b). Similarly, these two methods enhanced the partial factor productivity of nitrogen (PFPN), increasing it by 6.8% under drip irrigation and by 5.5% under micro-sprinkler irrigation (Figure 2c). Notably, alternate root-zone irrigation also showed a marked improvement in WUE, with an average increase of 116.2%, despite being supported by a limited number of observations (Figure 2b).

In contrast, other irrigation strategies such as mobile sprinkler irrigation, furrow irrigation, and subsurface sprinkler irrigation did not show statistically significant effects on yield, WUE, or PFPN. This lack of effect is likely due to small sample sizes and high variability, which contributed to wide uncertainty in the results. Drip irrigation and micro-sprinkler irrigation together represented more than 97% of all observations in the dataset, accounting for 71.9% and 25.1%, respectively (Figure 2d). This highlights their central role in current water-saving irrigation research.

### 2.3. Effects of Irrigation Levels on Wheat Yield, WUE, and PFPN Under Water-Saving Practices

Water-saving irrigation (WSI) practices at different irrigation levels had variable effects on wheat yield, water use efficiency (WUE), and partial factor productivity of nitrogen (PFPN) compared to full irrigation (100% FI) (Figure 3). Overall, wheat yield increased significantly under both >100% FI and 100% FI treatments, with average effect sizes of +8.9% and +9.2%, respectively. However, at 80–100% FI and 60–80% FI, yield showed no significant change. As irrigation levels decreased below 60% FI, yield began to decline. Notably, at 0–20% FI, yield decreased significantly by –18.4%, indicating that severe deficit irrigation severely reduces wheat productivity (Figure 3a).

WUE increased significantly under all irrigation levels ranging from 20% to 100% of full irrigation, while minimal or no improvement was observed under >100% FI and 0–20% FI. Similarly, PFPN improved across most irrigation levels, particularly between 40% and >100% FI, but showed little to no improvement when irrigation was below 40% FI. A similar trend was also observed under both drip and micro-sprinkler irrigation systems (Figure 3).

### 2.4. Differential Responses of Yield, WUE, and PFPN to WSI Under Varying Soil Conditions

WSI practices had varied effects on wheat yield, water use efficiency (WUE), and partial factor productivity of nitrogen (PFPN) across different soil bulk densities and soil textures (Figure 4). When soil bulk density exceeded 1.35 g/cm^3^, WSI practices showed no significant effect on yield, but led to notable increases in WUE (+15.0%) and PFPN (+6.0%). At a bulk density of ≤1.35 g/cm^3^, yield and PFPN both showed non-significant improvements (+3.7% and +3.6%, respectively), while WUE increased moderately (+32.5%). Regarding soil texture, the greatest yield improvements were observed in silt loam (+6.0%) and loam (+2.7%) soils. WSI practices had no significant effect on yield in clay loam, and reduced yield in sandy loam (−4.9%). For PFPN, the most pronounced gains were observed in loam (+14.5%) and silt loam (+9.4%), while no improvement was found in clay loam (+0.2%). Similarly, water use efficiency (WUE) was also notably enhanced in loam soils (+21.4%), indicating that medium-textured soils are more favorable for improving water productivity under WSI practices (Figure 4a–c).

Under drip irrigation, WSI practices had no significant effect on yield in clay loam, and reduced yield in sandy loam (−7.2%), but significantly improved WUE and PFPN in loam and silt loam soils. PFPN increased by 10.5% in loam and 16.5% in silt loam, while remaining unchanged in clay loam (Figure 4d–f). For micro-sprinkler irrigation, no data were available for clay loam soils. Yield improvements were observed in sandy loam (+3.3%), loam (+4.3%), and silt loam (+5.8%) soils. In loam soils, micro-sprinkler irrigation also enhanced WUE and PFPN by 4.1% and 6.1%, respectively (Figure 4g–i).

### 2.5. Meta-Regression Analysis of Climatic Factors Affecting Yield, WUE, and PFPN Under WSI

Meta-regression analyses revealed that growing season precipitation (GSP) was significantly negatively correlated with the effect sizes for yield and water use efficiency (WUE) (Figure 5a,b; *p* < 0.001). In contrast, no significant relationship was observed between GSP and partial factor productivity of nitrogen (PFPN) (Figure 5c; *p* = 0.3329). Mean annual temperature (MAT) showed no significant correlation with the effect sizes for yield, WUE, or PFPN (Figure 5d–f).

## 3. Discussion

### 3.1. Optimized Irrigation Achieves Resource Savings with No Yield Penalty

This meta-analysis demonstrates that optimized irrigation practices—particularly drip and micro-sprinkler systems—can significantly reduce both water and nitrogen inputs without compromising wheat yield (Figure 1). On average, irrigation inputs decreased by 35.1%, while nitrogen inputs declined by 7.2%, consistent with previous findings that precision irrigation reduces resource use through targeted delivery and reduced losses via evaporation and leaching [7,20]. Notably, these reductions were not accompanied by yield losses, suggesting that the conventional full irrigation and fertilization practices may represent excessive input levels in many parts of China’s wheat-growing regions. This aligns with recent shifts in agronomic research from yield maximization toward resource use efficiency and environmental sustainability [24].

### 3.2. WSI Methods Differ in Effectiveness, with Drip and Micro-Sprinkler Irrigation Leading Current Practice

Among the various water-saving irrigation (WSI) techniques evaluated, drip and micro-sprinkler irrigation emerged as the most consistently effective methods, particularly in improving water use efficiency (WUE) and partial factor productivity of nitrogen (PFPN), albeit with relatively modest yield gains. Specifically, drip irrigation increased WUE by 18.7% and PFPN by 6.8%, while micro-sprinkler irrigation produced comparable improvements, highlighting their broad adaptability and agronomic effectiveness under diverse agroecological conditions (Figure 2). These advantages are largely attributable to underlying physiological mechanisms, such as enhanced root-zone moisture distribution and reduced evapotranspiration losses [25].

Despite the limited yield responses observed (4.1% for micro-sprinkler and 0.3% for drip irrigation), the primary benefits of these WSI methods lie in resource use efficiency improvements rather than yield enhancement per se, particularly under well-managed production systems. Drip and micro-sprinkler irrigation together accounted for more than 97% of the observations in the dataset, reinforcing their status as the mainstream technologies currently adopted in water-saving agricultural practices, particularly in northern China.

Given that drip and micro-sprinkler together comprise ~97% of all observations, while alternate root-zone irrigation (ARI) is represented by only a few effect sizes, cross-method contrasts—particularly those involving ARI—are subject to wider uncertainty and possible small-sample/selection bias. We therefore interpret ARI estimates as provisional, emphasize drip and micro-sprinkler as the most evidence-supported options at present, and report method-specific sample sizes alongside effect estimates for transparency. The apparent WUE advantage of ARI should be validated by multi-site trials before broad recommendation.

Alternate root-zone irrigation demonstrated the highest WUE gain (116.2%), suggesting substantial potential. However, its limited representation in the dataset calls for further empirical validation across multiple sites and cropping systems before broader recommendation. By contrast, furrow irrigation, mobile sprinkler systems, and subsurface irrigation showed inconsistent or negligible effects on both efficiency and yield-related metrics, likely due to high site-specific variability and insufficient empirical evidence. These disparities underscore the need for targeted technology selection based on local context and data-informed decision-making.

In light of these findings, future policies and regional investment strategies should prioritize drip and micro-sprinkler irrigation as cornerstone technologies for irrigation modernization. Their proven effectiveness, widespread adoption, and strong empirical support render them critical components of sustainable water-saving strategies in northern China’s agricultural development.

### 3.3. Moderate Deficit Irrigation Optimizes WUE and PFPN While Avoiding Yield Loss

Our results demonstrate that WSI effectiveness is highly dependent on irrigation levels. Full irrigation (100% FI) and slightly excessive irrigation (>100% FI) significantly increased yield, whereas moderate deficit irrigation (60–100% FI) led to non-significant yield changes but substantially improved WUE and PFPN (Figure 3). Notably, extreme deficit irrigation (<40% FI) resulted in strong yield penalties (up to −18.4% at 0–20% FI), echoing findings that wheat has limited tolerance to drought during critical growth stages such as flowering and grain filling. Importantly, WUE peaked under moderate water stress conditions (40–80% FI), consistent with the principle that slight water stress can trigger physiological adjustments, such as enhanced root growth or improved stomatal regulation, without compromising productivity [26].

These insights highlight the potential for fine-tuned deficit irrigation strategies that maintain yield while maximizing resource efficiency. However, they also underscore the need for careful management to avoid crossing the threshold into yield-limiting water stress, particularly under increasing climate variability.

### 3.4. Soil Properties Modulate the Agronomic Benefits of WSI

Our meta-analysis reveals significant interactions between soil characteristics and the agronomic outcomes of WSI. For instance, medium-textured soils (silt loam and loam) consistently showed improvements in yield, WUE, and PFPN, whereas clay loam soils showed limited responsiveness and sandy loam even displayed yield reductions (Figure 4). These findings suggest that soil water-holding capacity and nutrient retention play pivotal roles in determining the success of WSI practices. In clay loam soils, high bulk density and poor infiltration may constrain the effectiveness of drip or sprinkler systems, while in sandy soils, rapid drainage may limit water availability and nutrient retention in the root zone [27]. Additionally, WSI significantly improved WUE and PFPN under high soil bulk density (>1.35 g/cm^3^), suggesting its effectiveness in compacted soils and the potential benefits of combining WSI with soil structure-enhancing practices.

Irrigation water quality and groundwater depth were not included in this meta-analysis because of limited and inconsistent reporting across primary studies. These factors, however, may substantially influence WSI outcomes—for example, saline irrigation water under deficit regimes can exacerbate root-zone salt accumulation, and shallow water tables in fine-textured soils may intensify secondary salinization [28,29]. Future research should therefore incorporate these variables to provide a more comprehensive assessment of WSI performance.

### 3.5. Climatic Conditions Shape WSI Performance

Climate conditions, particularly growing season precipitation (GSP), significantly influenced WSI outcomes (Figure 5). We observed a negative relationship between GSP and the effect sizes for yield and WUE, indicating that the benefits of WSI are most pronounced under drier conditions where irrigation plays a larger role in yield determination. This finding aligns with previous regional studies suggesting that irrigation efficiency gains are maximized in water-limited environments [30,31]. In contrast, no significant relationship was found between GSP and PFPN, implying that nitrogen use efficiency under WSI may be less sensitive to rainfall variability than water use efficiency.

Interestingly, mean annual temperature did not significantly affect WSI outcomes, suggesting that within the temperature range observed in China’s wheat-growing regions, thermal conditions are not a major limiting factor. However, rising temperatures in future scenarios may interact with water and nutrient dynamics in complex ways, calling for more research on WSI performance under climate change projections.

### 3.6. Policy Implications for Sustainable Wheat Production in China

The findings of this meta-analysis provide compelling evidence to support the expansion of optimized water-saving irrigation (WSI) practices—particularly drip and micro-sprinkler systems—as part of national strategies for agricultural water and nutrient conservation. Given China’s growing water scarcity and the critical role of the North China Plain in wheat production, scaling up WSI could serve dual goals of ensuring food security and promoting resource efficiency. Policymakers should consider integrating WSI adoption into the National Farmland Irrigation Development Plan and the Black Soil Protection Initiative, with targeted subsidies, cost-sharing programs, or performance-based incentives to promote farmer adoption, particularly in areas with high evapotranspiration and limited precipitation. Moreover, given the observed trade-off between severe water savings and yield loss under extreme deficit irrigation, technical guidance and irrigation scheduling tools (e.g., soil moisture sensors, real-time decision support systems) should be prioritized to avoid under-irrigation. Capacity-building efforts, including farmer training and extension services, will be essential for translating the benefits of WSI from experimental settings to large-scale on-farm application.

## 4. Materials and Methods

### 4.1. Data Collection

A systematic literature search was conducted to evaluate the effects of water-saving irrigation (WSI) practices on wheat grain yield, water use efficiency (WUE), and partial factor productivity of nitrogen (PFPN) in China. Following the PRISMA (Preferred Reporting Items for Systematic Reviews and Meta-Analyses) guidelines (Figure 6), peer-reviewed publications from 2000 to June 2025 were retrieved from the Web of Science and China National Knowledge Infrastructure (CNKI) databases using combinations of Chinese and English keywords such as “wheat”, “water-saving irrigation”, “regulated deficit irrigation”, “drip irrigation”, “micro-sprinkler irrigation”, “micro-irrigation”, “grain yield”, and “water use efficiency”. The search included the title, abstract, and keyword fields, using Boolean syntax (e.g., TI = (“drip irrigation”) OR AB = (“micro-sprinkler irrigation”)).

Studies were included based on the following criteria: (1) field experiments conducted in mainland China; (2) winter wheat as the target crop, with at least one WSI treatment and one conventional or full irrigation control; (3) provision of at least one outcome variable among yield, WUE, and PFPN; and (4) availability of sufficient statistical information such as standard deviation (SD), standard error (SE), or sample size (n). When multiple sites or years were reported within a study, each was treated as an independent observation.

From each eligible publication, the following data were extracted: (1) general information, including author name, publication year, and study location; (2) experimental details such as irrigation method (e.g., drip, micro-sprinkler, sprinkler, or alternate root-zone irrigation), irrigation timing and amount, and nitrogen application rate; (3) soil type, bulk density, seasonal precipitation, and mean growing-season temperature; and (4) outcome variables including grain yield, evapotranspiration, and WUE. All data were digitized using a standardized data extraction template. For studies presenting data only in graphical format, numerical values were extracted using WebPlotDigitizer (version 3.4). Outliers were removed from the dataset prior to analysis. The geographic distribution of the 48 included study sites is shown in Figure 7.

### 4.2. Meta-Analysis Procedure

To quantitatively assess the impacts of water-saving irrigation (WSI) practices on wheat grain yield, water use efficiency (WUE), and partial factor productivity of nitrogen (PFPN) in China, we employed a natural logarithmic response ratio (lnR), which is widely adopted in ecological and agricultural meta-analyses, as the effect size metric:(1)$$lnR=\ln\left(X_{t}/X_{c}\right)$$
Here X_t_ and X_c_ denote the means of the treatment (water-saving irrigation) and control (conventional irrigation) groups, respectively.

The variance (v) of lnR for each observation was calculated as follows:(2)$$v={{\mathrm{SD}}_{t}}^{2}/n_{t}{X_{t}}^{2}+{{SD}_{c}}^{2}/n_{c}{X_{c}}^{2}$$
Here, SD_t_, SD_c_, n_t_, and n_c_ represent the standard deviations and replicate numbers of the treatment and control groups, respectively.

The weighted effect sizes (lnR_++_) were determined by the following equation:(3)$${lnR}_{++}=\sum ({lnR}_{i}{\;W}_{i})/\sum \;(W_{i})$$
Here, lnR_i_ is the effect size of the *i*-th comparison and W_i_ is the corresponding weight, defined as follows:(4)$$W_{i}\left.=1/(v_{i}+\tau^{2}\right)$$
Here, v_i_ denotes the sampling variance of lnR_i_, and τ^2^ is the between-study variance, estimated using the restricted maximum likelihood (REML) method in the rma.mv function of the R package “metafor” (version 4.5.1), implemented in R version 4.5.1.

The standard error of the weighted mean and its 95% confidence interval (CI) were calculated as follows:(5)$$S_{lnR++}=\sqrt{1/\sum W_{i}}$$95% CI = lnR_++_ ± 1.96 S_lnR++_
(6)

For studies that did not report standard deviations (SDs), the impute_SD function in the ‘metagear’ package (R version 4.5.1) was employed to impute missing values [32]. To account for the non-independence of effect sizes arising from shared control groups, a variance–covariance matrix was constructed following the method of Lajeunesse [33].

A random-effects model was employed; this modeled variance at the study level, treatment level, and sampling error level. Parameter estimation was conducted using restricted maximum likelihood (REML). The ‘rma’ function in the ‘metafor’ package (R version 4.5.1) was used to estimate the weighted effect sizes (lnR_++_) and their 95% confidence intervals (CIs). An effect was deemed statistically significant if the CI did not overlap with zero. Between-group differences were considered significant if the respective CIs did not overlap [34]. Heterogeneity among studies was assessed using the Q statistic (Qt), with statistical significance indicating variation in effect sizes potentially attributable to moderator variables (see Appendix A) [35]. Subgroup analyses were conducted for categorical moderators with at least ten observations or at least five observations from two or more independent studies [36]. Meta-regression analyses were conducted using the “rma()” function with the restricted maximum-likelihood estimator (REML) in the “metafor” package (R version 4.5.1) to examine the relationships between effect sizes and climatic variables, including growing season precipitation (GSP) and mean annual temperature (MAT) [37].

To enhance interpretability, effect sizes were transformed into percentage changes using the following equation:(7)$$Percent\;change=\left({exp}^{lnR}-1\right)\;100\%$$

### 4.3. Model Diagnostics

The normality of effect size distributions was assessed using quantile–quantile (Q–Q) plots. As shown in Appendix A, the plotted points aligned closely with the 1:1 line, indicating an approximately normal distribution of lnR values and supporting the suitability of parametric meta-analysis models.

Publication bias was evaluated using Rosenthal’s fail-safe number. For each soil indicator, the calculated fail-safe number exceeded the threshold of 5k + 10, where *k* was the number of effect sizes (Appendix A), suggesting that a substantial number of unpublished null-result studies would be required to overturn the significance of our findings. These results confirmed that the meta-analysis outcomes were robust and not substantially affected by potential publication bias.

## 5. Conclusions

This meta-analysis demonstrates that optimized water-saving irrigation (WSI), especially drip and micro-sprinkler systems, can significantly reduce water (–35.1%) and nitrogen inputs (–7.2%) without compromising wheat yield. These systems also enhance water use efficiency (WUE) and nitrogen productivity (PFPN), particularly under moderate deficit irrigation (60–100% FI) and in medium-textured soils. WSI remained effective under high soil bulk density (>1.35 g/cm^3^), and its benefits were most pronounced in low-precipitation regions. Among evaluated methods, drip and micro-sprinkler irrigation—accounting for over 97% of the observations—were the most reliable and broadly applicable technologies. Therefore, they should be prioritized in future irrigation modernization and resource conservation policies, especially in water-scarce areas such as northern China. Successful scaling of WSI will require not only financial incentives but also site-specific management tools and capacity-building efforts.

## Figures and Tables

**Figure 1 plants-14-02837-f001:**
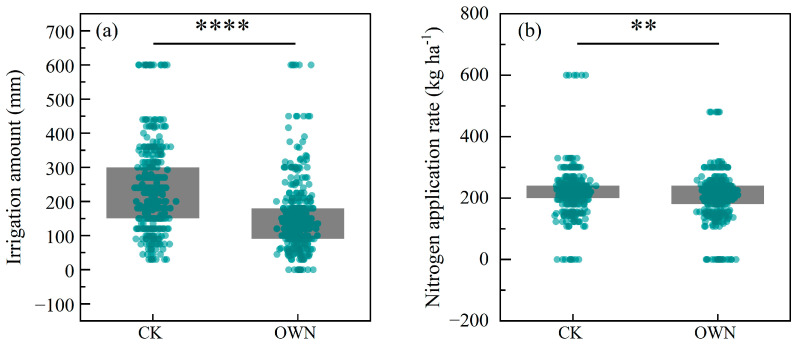
Comparison of (**a**) irrigation amount and (**b**) nitrogen application rate between the conventional treatment (CK) and the optimized water and nitrogen management treatment without yield reduction (OWN). Gray boxes represent the interquartile range with median values; dots represent individual data points from included studies. Significant reductions in both irrigation and nitrogen inputs were observed under the OWN treatment (** *p* < 0.01; **** *p* < 0.0001, unpaired *t* test).

**Figure 2 plants-14-02837-f002:**
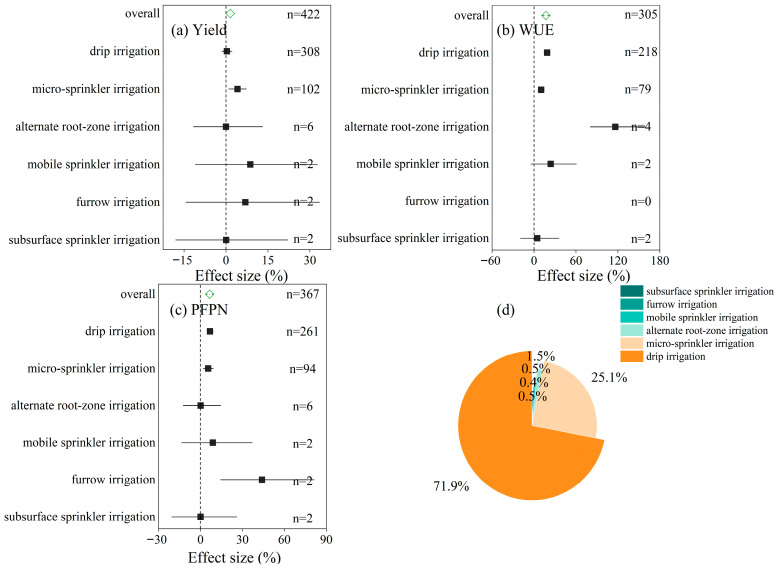
Effects of different water-saving irrigation methods on (**a**) wheat yield, (**b**) water use efficiency (WUE), and (**c**) partial factor productivity of nitrogen (PFPN), based on a meta-analysis. Effect sizes (%) are shown with 95% confidence intervals. Positive values indicate an increase relative to the conventional treatment. The vertical dashed line represents no effect (effect size = 0). (**d**) Proportion of observations for each irrigation method included in the dataset.

**Figure 3 plants-14-02837-f003:**
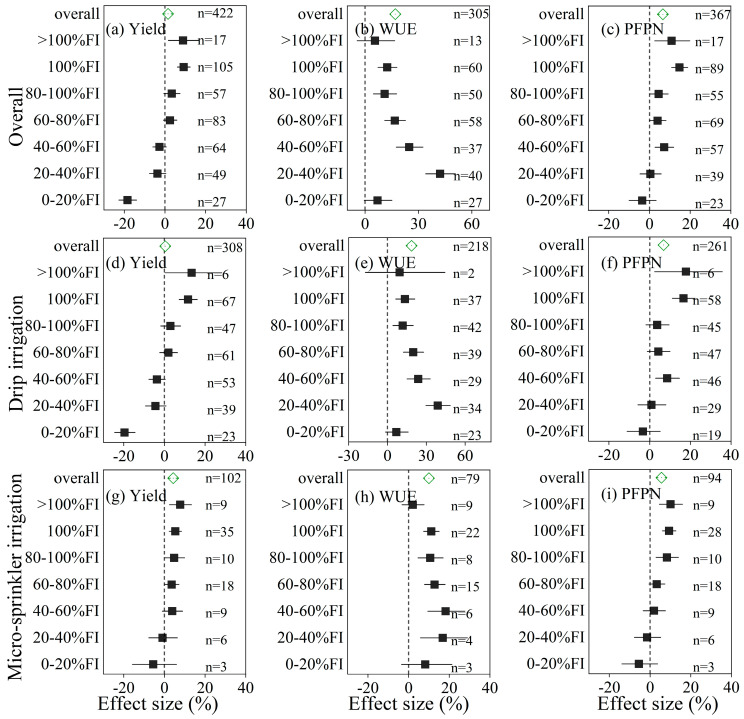
Effect sizes (%) of different irrigation levels, expressed as percentages of full irrigation (FI), on wheat yield (**a**,**d**,**g**), water use efficiency (WUE) (**b**,**e**,**h**), and partial factor productivity of nitrogen (PFPN) (**c**,**f**,**i**). Results are shown for all irrigation methods combined (**a**–**c**), drip irrigation (**d**–**f**), and micro-sprinkler irrigation (**g**–**i**). Irrigation levels were grouped into: >100% FI, 100% FI, 80–100% FI, 60–80% FI, 40–60% FI, 20–40% FI, and 0–20% FI. Effect sizes represent changes relative to 100% FI (reference group), with horizontal lines indicating the 95% confidence intervals. Positive values indicate improvement compared to full irrigation, and vertical dashed lines denote no effect (0%). Sample sizes (n) are indicated for each subgroup.

**Figure 4 plants-14-02837-f004:**
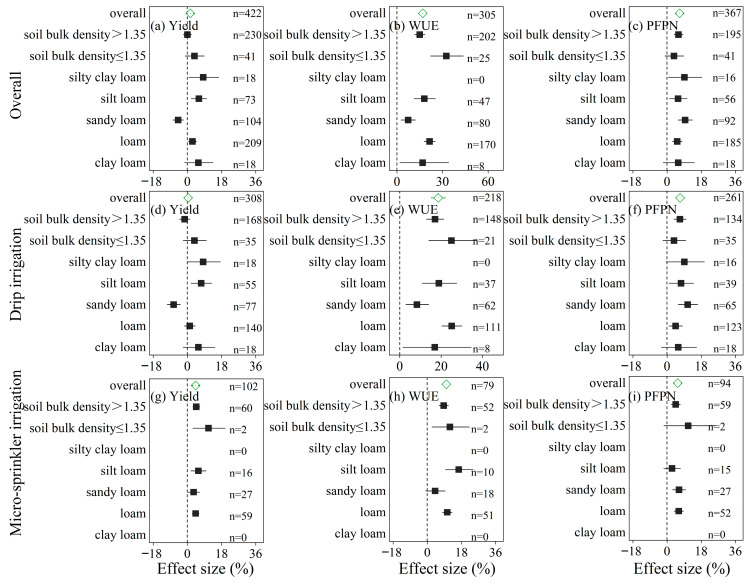
Effect sizes (%) of water-saving irrigation (WSI) practices on wheat yield (**a**,**d**,**g**), water use efficiency (WUE) (**b**,**e**,**h**), and partial factor productivity of nitrogen (PFPN) (**c**,**f**,**i**) under different soil bulk densities and soil texture types. Results are presented for all data combined (**a**–**c**), drip irrigation (**d**–**f**), and micro-sprinkler irrigation (**g**–**i**). Effect sizes are calculated relative to conventional irrigation, with horizontal bars representing 95% confidence intervals. The vertical dashed line indicates no effect (0%), and sample size (n) is shown for each subgroup.

**Figure 5 plants-14-02837-f005:**
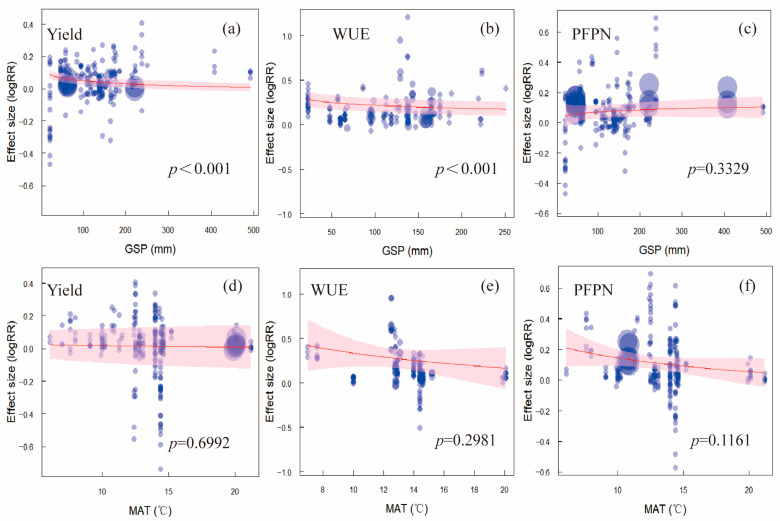
Meta-regression analysis of the effects of growing season precipitation (GSP) and mean annual temperature (MAT) on yield, water use efficiency (WUE), and partial factor productivity of nitrogen (PFPN) under water-saving irrigation practices. The red line represents the fitted effect size curve, while the blue circles indicate individual effect sizes. The shaded area around the red line represents the confidence interval. (**a**) Yield vs. GSP; (**b**) WUE vs. GSP; (**c**) PFPN vs. GSP; (**d**) Yield vs. MAT; (**e**) WUE vs. MAT; (**f**) PFPN vs. MAT.

**Figure 6 plants-14-02837-f006:**
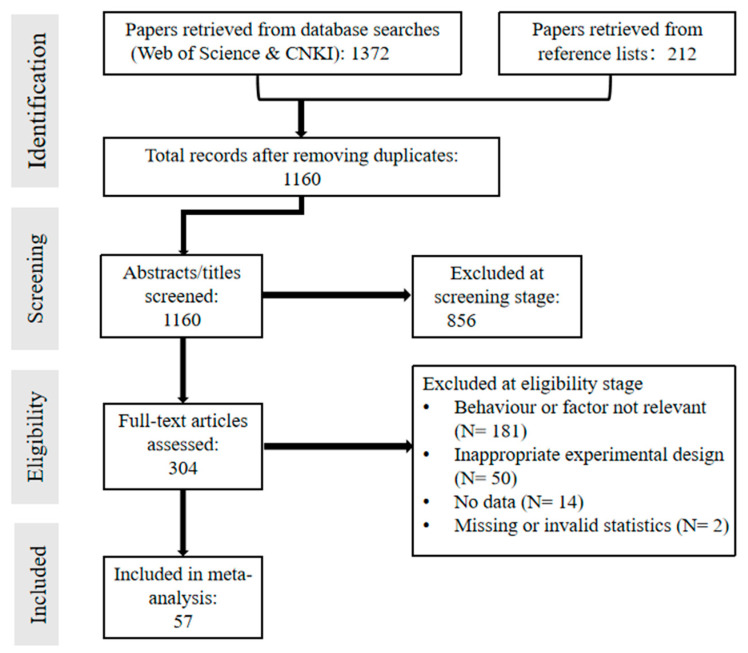
Preferred reporting items for systematic reviews and meta-analyses (PRISMA) flow diagram for manuscript selection.

**Figure 7 plants-14-02837-f007:**
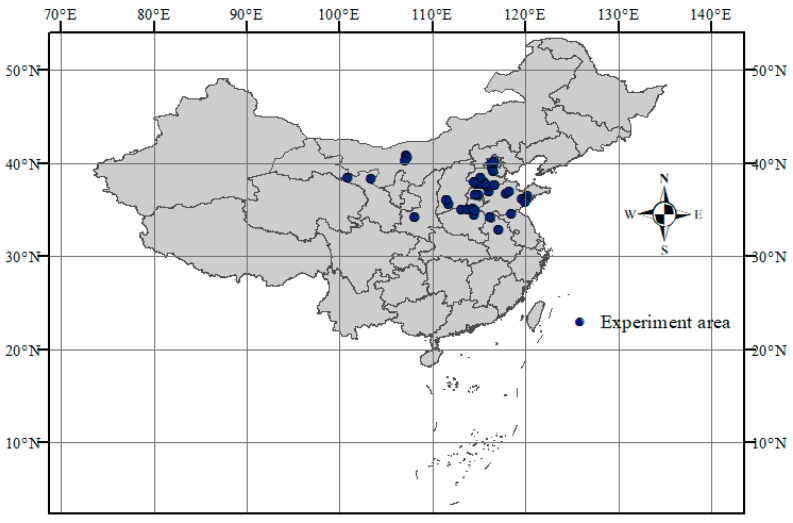
Geographic distribution of cover cropping experiments across 47 independent study sites included in the meta-analysis.

## Data Availability

Data is contained within the article and Appendix A.

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
