# Peer review of "Evaluating the Effectiveness of Water-Saving Irrigation on Wheat (Triticum aestivum L.) Production in China: A Meta-Analytical Approach"

_plants, 2025, doi:10.3390/plants14182837_

Round 1

Reviewer 1 Report

Comments and Suggestions for Authors

The Introduction is well written, with clearly stated objectives. However, the role of nitrogen is not sufficiently integrated into the main narrative. The section would benefit from a more thorough discussion of nitrogen dynamics and its interactions with irrigation practices, as this is central to understanding water–nutrient synergies in wheat production.

The Results are presented clearly and logically, with appropriate use of figures and tables.

The Discussion is well structured, concise, and provides sound conclusions. It effectively synthesizes the findings and places them in the context of existing literature. However, it should be noted that drip and micro-sprinkler irrigation account for 97% of the dataset. This raises the question of whether the comparisons with other irrigation methods, particularly alternate root-zone irrigation—which showed the largest improvements in water use efficiency (WUE)—may be biased. The limited representation of alternate root-zone irrigation in the dataset suggests that its potential benefits could be underrepresented, and the discussion would benefit from explicitly acknowledging this limitation. 

Reviewer 2 Report

Comments and Suggestions for Authors

Dear Authors, I don't have any recommendations or suggestions.  Just I suggest to try to improve figure quality for a clearer reading. As an example, in Fig. 3 and 4  can be useful to use the same x scale for each parameter (e.g., Fig. 3, yield from -20 to +40 for each irrigation methods).

Kind regards

Reviewer 3 Report

Comments and Suggestions for Authors

Reviewer 4 Report

Comments and Suggestions for Authors

Line 101: if a set of results is not statistically significant, it should not be considered.

Line 291: Fig. 1 should be numbered Fig 6 along the text.

Line 315: The same for Fig 2, which should be Fig. 7.

Line 348- “Heterogeneity among studies was assessed using the Q statistic (Qt)”, but the Q statistic is not shown in the main text or the SM.

Graphs do not show their statistics regarding the significance of the effect.

The "raw" table with the full dataset used in the meta-analysis is not presented.
